# Differential Effects of Pancreatic Cancer-Derived Extracellular Vesicles Driving a Suppressive Environment

**DOI:** 10.3390/ijms241914652

**Published:** 2023-09-27

**Authors:** Anurag Purushothaman, Jacqueline Oliva-Ramírez, Warapen Treekitkarnmongkol, Deivendran Sankaran, Mark W. Hurd, Nagireddy Putluri, Anirban Maitra, Cara Haymaker, Subrata Sen

**Affiliations:** 1Department of Translational Molecular Pathology, MD Anderson Cancer Center, Houston, TX 77030, USA; apurushothaman@mdanderson.org (A.P.); jloliva@mdanderson.org (J.O.-R.); wtreekit@mdanderson.org (W.T.); deivendran.kumar@gmail.com (D.S.); amaitra@mdanderson.org (A.M.); 2Department of Thoracic/Head and Neck Medical Oncology, MD Anderson Cancer Center, Houston, TX 77030, USA; 3Ahmed Center for Pancreatic Cancer Research, MD Anderson Cancer Center, Houston, TX 77030, USA; mwhurd@mdanderson.org; 4Department of Molecular and Cellular Biology, Baylor College of Medicine, Houston, TX 77030, USA; putluri@bcm.edu; 5Dan L Duncan Cancer Center, Advanced Technology Core, Alkek Center for Molecular Discovery, Baylor College of Medicine, Houston, TX 77030, USA; 6Department of Pathology, MD Anderson Cancer Center, Houston, TX 77030, USA

**Keywords:** pancreatic ductal adenocarcinoma, extracellular vesicles, hyaluronan, monocytes, hexosamine biosynthetic pathway

## Abstract

Pancreatic ductal adenocarcinoma (PDAC) cells display extensive crosstalk with their surrounding environment to regulate tumor growth, immune evasion, and metastasis. Recent advances have attributed many of these interactions to intercellular communication mediated by small extracellular vesicles (sEVs), involving cancer-associated fibroblasts (CAF). To explore the impact of sEVs on monocyte lineage transition as well as the expression of checkpoint receptors and activation markers, peripheral blood monocytes from healthy subjects were exposed to PDAC-derived sEVs. Additionally, to analyze the role of sEV-associated HA in immune regulation and tissue-resident fibroblasts, monocytes and pancreatic stellate cells were cultured in the presence of PDAC sEVs with or depleted of HA. Exposure of monocytes to sEVs resulted in unique phenotypic changes in HLA-DR, PD-L1, CD86 and CD64 expression, and cytokine secretion that was HA-independent except for IL-1β and MIP1β. In contrast, monocyte suppression of autologous T cell proliferation was reduced following exposure to HA-low sEVs. In addition, exposure of stellate cells to sEVs upregulated the secretion of various cytokines, including MMP-9, while removal of HA from PDAC-derived sEVs attenuated the secretion of MMP-9, demonstrating the role of sEV-associated HA in regulating expression of this pro-tumorigenic cytokine from stellate cells. This observation lends credence to the findings from the TCGA database that PDAC patients with high levels of enzymes in the HA synthesis pathway had worse survival rates compared with patients having low expression of these enzymes. PDAC-derived sEVs have an immune modulatory role affecting the activation state of monocyte subtypes. However, sEV-associated HA does not affect monocyte phenotype but alters cytokine secretion and suppression of autologous T cell proliferation and induces secretion of pro-tumorigenic factors by pancreatic stellate cells (PSC), as has been seen following the conversion of PSCs to cancer-associated fibroblasts (CAFs). Interruption of the hexosamine biosynthetic pathway, activated in PDAC producing the key substrate (UDP-GlcNAc) for HA synthesis, thus, represents a potential clinical interception strategy for PDAC patients. Findings warrant further investigations of underlying mechanisms involving larger sample cohorts.

## 1. Introduction

Pancreatic ductal adenocarcinoma (PDAC) has one of the highest death rates of any solid tumor with a 5-year survival rate of about 10% [1]. Despite improvements in therapeutic management, the cure rates have increased only minimally, with PDAC currently being the third most common cancer-related death in the United States [2]. PDAC is characterized by an extremely rich desmoplastic stroma deposition, accompanied by profound immune suppression and hypovascularization [3]. An in-depth understanding of the fundamental mechanisms of tumor and immune cell communication is necessary for developing new therapeutic strategies to effectively target this disease. Cell-derived extracellular vehicles (EVs), especially small extracellular vesicles (sEVs) (EVs smaller than 200 nm in diameter according to MISEV2018 [4]), are central mediators of intercellular communications. The International Society for Extracellular Vesicles (ISEV) proposed Minimal Information for Studies of Extracellular Vesicles (“MISEV”) guidelines for the field in 2014 and updated the guidelines in MISEV2018. Several studies over the last decade have shown that sEVs derived from pancreatic cancer cells make use of their molecular and genetic cargo to reprogram recipient cells to promote tumor growth [5] and pre-metastatic niche construction [6], and enhance angiogenesis [7], immune regulation [8,9,10], and therapeutic resistance [11].

An immunosuppressive and pro-tumorigenic phenotype observed in the PDAC tumor microenvironment is attributed to the imbalance in the number of immune cells, such as immunosuppressive T regulatory cells (Treg), M2 polarized tumor-associated macrophages, and myeloid-derived suppressor cells (MDSCs) that prevail over cytotoxic CD8 T cells, and dendritic cells. MDSCs and M2 populations in PDAC also contribute to the secretion of arginase, nitric oxide synthase, IL-10, and TGFβ, which drives tumor invasion/growth and inhibition of antitumoral responses from NK and T cells [12]. In pancreatic cancer, elevated monocytes, which are crucial for innate inflammatory responses, are shown to have an immunosuppressive phenotype by downregulation of HLA-DR (MHC class two receptor molecule, encoded on the gene human leukocyte antigen complex) and lower phosphorylation levels of main transcription factors like STAT1 and NFkB compared with healthy donors [8,13]. There are three subtypes of monocytes based on the expression of surface markers, morphology, and functionality: classical (CD14high, CD16−), intermediate (CD14high, CD16low), and nonclassical (CD14low, CD16high). Tumor-induced systemic environment influences both the frequency of monocytes and their phenotypes to acquire an immunosuppressive activity, primarily by altering the expression of cell surface antigen-presenting proteins and checkpoint receptors [14,15]. To a certain extent, this imbalance in immune cell activity within the tumor microenvironment is mediated through tumor-derived EVs. For example, in melanoma and colon cancer, tumor-derived exosomes block the differentiation of peripheral inflammatory monocytes to dendritic cells and favor their polarization to the monocytic-MDSC phenotype, which is characterized by decreased expression of HLA-DR [16]. Moreover, exosomes derived from PDAC cells promote polarization to M2-like macrophages and secretion of pro-tumorigenic proteins, including VEGF, IL-6, MMP-9, and MCP-1 [9]. PDAC-derived EVs induce immune suppression in monocytes and the mechanisms involving an alteration in STAT3 signaling, downregulation of HLA-DR, induction of arginase expression, and generation of reactive oxygen species [8]. Despite these findings, there is no evidence of whether PDAC EVs could directly modulate monocyte subpopulations. Our study reveals a direct effect of PDAC cell-derived sEVs on monocyte-secreted pro- and anti-inflammatory and chemotactic proteins, unlike the sEVs derived from normal pancreatic ductal epithelial cells.

It has been demonstrated that interactions of PDAC EVs with monocytes and downstream activation of STAT3 signaling followed by the secretion of chemokines are mediated through the CD44 receptors on monocytes [10]. Blocking CD44 function on monocytes using an anti-CD44 mAb reduces the engulfment of PDAC-derived EVs and decreases activation of STAT3 signaling and levels of CCL4 and CCL5. The glycosaminoglycan hyaluronan (HA) is the major ligand for CD44. HA is synthesized at the plasma membrane, constituted of disaccharide glucuronic acid (GlcUA) repeats and N-acetylglucosamine (GlcNAc) to form large polysaccharides. HA has been shown to be abundantly accumulated in the surrounding stroma of malignant tumors, and HA-rich microenvironment promotes tumor progression by enhancing tumor-associated macrophage recruitment, migration/invasion, metastasis/angiogenesis, and drug resistance [17]. Pancreatic cancer cells secrete HA, and the tumors are characterized by a dense desmoplastic stroma, associated with the highest incidence of detectable HA content compared with any other malignant tumors [18]. We report for the first time that PDAC-derived sEVs carry HA. The hexosamine biosynthetic pathway (HBP, a shunt pathway of glycolysis), which is overactivated in PDAC [19], synthesizes one of the primary substrates (UDP-GlcNAc) for HA biosynthesis. HA stimulates pancreatic cancer cell mobility and abnormal accumulation of HA elevates the interstitial pressure within the PDAC tumor microenvironment, posing a barrier to the delivery of chemotherapeutic agents [20], thus correlating with worsened prognosis in PADC patients [18]. However, it remains unclear if HA could promote changes in the immune profile of monocyte subpopulations and cytokine secretion by pancreatic stellate cells. Our novel findings on the presence of HA in PDAC sEVs reveal the involvement of the PDAC sEVs in modulating the immune landscape with significant implications for developing effective immunotherapy for PDAC.

Pancreatic stellate cells (PSCs) express CAF-like surface markers such as α-SMA upon activation, converting them to activated CAFs. PSC-derived CAFs contribute to tumor stroma in PDAC by secreting aberrant extracellular matrix proteins, matrix metalloproteinases (MMPs), and tissue inhibitors of matrix metalloproteinases (TIMPs), which promote desmoplasia [21]. Extensive desmoplasia. which is remarkably characterized as a hallmark of the PDAC microenvironment, promotes tumor growth, cell invasion, and chemoresistance [22]. During PDAC progression, PDAC can activate PSC cell proliferation and production of their matrix synthetic components and enzymes [23]. However, the impact of PDAC sEVs on PSCs have not been well elucidated.

In this study, we demonstrate that the PDAC sEVs play roles in the differentiation of monocytes and subset-specific expression of activation and inhibitory receptors as well as antigen presentation machinery. Additionally, the results show that HA presence in sEVs enhances monocyte production of IL-1β, influences the ability of monocytes to suppress T cell proliferation, and enhances stellate cell production of matrix metalloproteinase-9 (MMP-9), a feature associated with CAFs promoting tumor invasion and angiogenesis (Appendix A).

## 2. Results

### 2.1. Pancreatic Ductal Adenocarcinoma-Derived Small Extracellular Vesicles Preferentially Modulate the Monocyte Landscape

Aberrant monocyte function and subsequent immune suppression are associated with pancreatic cancer progression [13]. To determine if pancreatic cancer cell-derived EVs play a role in modulating the monocyte landscape, we isolated sEVs from different PDAC cell lines. sEVs were isolated from a medium conditioned by PDAC cells for 48 h and quantified by transmission electron microscopy and nanoparticle tracking analysis. As shown in Figure 1A,B, PDAC cell-derived sEVs presented a round shape morphology with a median vesicle size of 160 nm. Western blot analysis (Figure 1C) showed expression of exosome-associated sEV markers CD63 and ALIX, further confirming the fidelity of the EV preparations. To determine if sEV secretion by PDAC cells were altered by chemotherapeutic drugs, human PDAC cells were exposed to drugs such as gemcitabine and paclitaxel (Appendix A). These drugs are used as a standard first-line treatment for patients with metastatic PDAC. Human PDAC cell lines PANC-1 and AsPC-1 were exposed to the drug for 16 h, at which time approximately 40% of the cells treated with gemcitabine or paclitaxel underwent apoptosis and/or cell death. Analysis of the sEVs isolated from the conditioned media by these cells revealed that the number of EVs secreted by the tumor cells, as determined by ZetaView analysis, increased with the drug treatment (Appendix A). Plots of ZetaView analysis and transmission electron microscope images revealed that the particle size ranged between 160 to 180 nm (Appendix A).

To explore the role of sEVs in monocyte differentiation and activation, enriched monocytes from healthy normal donors were treated with sEVs overnight. We found by flow cytometry that the proportion of classical monocytes increased, while intermediate monocytes decreased in the presence of PDAC sEVs (Figure 1D, Gating Strategy Appendix A). In addition, there was a significant increase in HLA-DR and PD-L1 MFI only within the classical subset, while the costimulatory molecule CD86 increased in both classical and intermediate populations after exposure to AsPC1 and PANC-1 sEVs (Figure 1E–G). However, while neither HLA-DR nor PD-L1 expression on the nonclassical subset was impacted, a downregulation in the expression of the IgG receptor CD64 following exposure to the sEVs was observed (Figure 1H). No changes were observed in PD-L1 cell percentages as shown in Appendix A. Interestingly, the co-expression of HLA-DR and PD-L1 was upregulated during the exposure to sEVs for both classical and intermediate monocytes (Figure 1I). Our data indicate that the sEVs derived from PDAC cell lines have differential impacts on the monocyte subpopulations, which might be related to their singular immunological function. Finally, to test the suppressive effect of sEVs from PDAC on monocyte functionality, we co-cultured autologous T cells with monocytes previously exposed to sEVs. Our results showed that the PDAC-derived sEVs, but not sEVs from non-tumorigenic normal HPDE or HPNE cells, inhibited the T cell proliferation (Figure 1J,K). Furthermore, monocytes from normal donors were co-cultured with sEVs derived from AsPC1 and PANC-1 cell lines, compared with no stimulation or sEVs derived from normal pancreatic cell lines, HPDE and HPNE. Multiple proinflammatory/immune recruitment (IL-1β, IL-6, IL-8, TNFα, IFNα, MIP-1β, CXCL12) and immune suppressive cytokines (IL-10) were found to be induced by tumor line-derived sEVs but not those derived from normal pancreatic cell lines (Appendix A). Together this data suggest that the PDAC sEVs imprint a suppressive phenotype on monocytes that impacts their ability to induce inflammatory cytokines and impairs the co-stimulatory function necessary for T cell proliferation.

### 2.2. PDAC-Derived sEVs Are Enriched with the Glycosaminoglycan Hyaluronan

Hyaluronan is secreted by human cultured pancreatic cancer lines, and increased expression of HA is found in PDAC compared with normal pancreas [24,25,26]. The HA receptor, CD44, is expressed by monocytes and has been shown to interact with PDAC-derived sEVs [10]. Importantly, extensive accumulation of hyaluronan correlates with worsened prognosis in PDAC patients [27]. Unlike other glycosaminoglycan, hyaluronan is synthesized on the cellular plasma membrane, and the newly synthesized chain extrudes across the membrane directly into the extracellular space. In order to investigate if the sEVs released by PDAC cells are enriched with HA, sEVs were isolated from the conditioned medium of PDAC cell lines and HA levels were quantified by ELISA. Interestingly, significant levels of HA were detected in sEVs released by human PDAC cells (Figure 2A). These findings were further confirmed using a dot blot assay where the presence of HA was detected on PDAC-derived sEVs, immobilized on a strip of nitrocellulose membrane, using hyaluronan binding protein (Figure 2B). To investigate if HA expression on sEVs could be impacted by treatment in patients with PDAC, we isolated sEVs from serum samples collected from 10 PDAC patients with matched collections before the start of treatment and after undergoing neoadjuvant radiotherapy, chemotherapy, and/or surgical resection. The clinicopathological characteristics of these patients are shown in Appendix A. sEVs were isolated using columns based on size exclusion chromatography and integrity was analyzed by transmission electron microscopy. As shown in Figure 2C, electron microscopy revealed typical round shape morphology and size distribution, consistent with that of sEVs. To further elucidate whether these sEVs were enriched with HA, sEVs were subjected to ELISA and dot blot analysis. Consistent with that of the PDAC-cell line-derived sEVs, all patient-derived sEVs were found to be enriched with HA (Figure 2D). In addition, as shown in Figure 2E and Appendix A, in seven patients treated with neoadjuvant chemotherapy, HA levels in sEVs were elevated in patients’ post-treatment as compared with their pretreatment concentration, regardless of tumor grade. As equal amounts of sEV proteins are used for HA analysis, there is no concern about the elevated levels of HA detected in post-treatment sEVs due to an increase in circulating sEVs over time.

### 2.3. Participation of the Hyaluronan Present in sEVs and Immune Modulation

A recent study reported that stromal hyaluronan accumulation is associated with low immune response and poor prognosis in pancreatic cancer [28]. To determine if HA in PDAC-derived sEVs plays a role in monocyte lineage transitions as well as expression of checkpoint receptors and activation, HA levels were depleted in sEVs using 4-methylumbelliferone (4-MU). 4-MU is a potent HA synthesis inhibitor and is shown to inhibit enhanced HA synthesis and cell migration in pancreatic cancer [29]. sEVs purified from cells treated with 0.5 mM 4-MU for 48 h had significantly lower levels (~60% reduction) of HA compared with sEVs from cells treated with DMSO control (Figure 3A). No marked effects on cell viability were noted following treatment with 4-MU. To elucidate the impact of sEV-HA in monocyte lineage transition and checkpoint receptor expression, peripheral blood monocytes were incubated with basal HA PDAC sEVs and HA-low PDAC sEVs overnight. Unexpectedly, there were no significant changes in the monocyte subpopulation proportions or phenotype when comparing PANC-1 and PANC-1 HA-low (Figure 3B–F). Also, no change in the percentage expression of PD-L1 was observed; however, the correlation between the co-expression of HLA-DR and PD-L1 MFI on the intermediate subpopulation decreased in significance when the sEVs contained low HA (Appendix A). Finally, we determined if monocyte function was affected by HA driving specific T cell proliferation suppression and cytokine secretion profile. Monocytes were co-cultured with autologous T cells previously exposed to sEVs derived from PANC-1 cell lines and PANC-1 HA-low cells. Interestingly, monocytes exposed to HA-low sEVs reversed the T cell proliferation suppression seen when HA was present in PANC-1 cells (Figure 3G). When monocytes were incubated with sEVs from PANC-1 and PANC-1 HA-low cells, while most cytokines were not affected by the presence of HA, IL-1β and MIP-1β showed an inverse association with HA presence appearing to drive production of IL-1β and suppress MIP-1β (Figure 3H and Appendix A). Our results might indicate that the presence of HA is crucial for the suppression effect of PDAC sEVs on monocytes.

Pancreatic stellate cells (PSC) are key stromal cells within the desmoplastic microenvironment, and several studies revealed a bidirectional interaction between PDAC cells and PSCs that plays a key role in the progression of PDAC [30]. Specific stimulation from PDAC cells leads to PSC activation and secretion of different paracrine stimulants and growth factors that support tumor development [31]. To verify whether sEVs released by PDAC cells induce the secretion of cytokines/growth factors by PSCs, a medium conditioned by PSCs in the presence or absence of PDAC sEVs was subjected to an antibody microarray. Only serum-free media was used in our experiments to avoid any interference from cytokines and growth factors present in the added serum. As shown in Figure 3I,J, 10 different cytokines were found to be significantly abundant in the cell culture medium of stellate cells treated with PDAC sEVs, compared with conditioned medium from cells that were not exposed to sEVs. In addition, analysis of conditioned medium from PSCs treated with HA-high sEVs showed a significant increase in MMP-9, compared with media from cells treated with HA-low sEVs (Figure 3J). This finding is important because macrophage-secreted MMP-9 promotes tumor growth by inducing mesenchymal transition in pancreatic cancer cells [32]. Our findings suggest an alternate mechanism that PDAC cells might use to trigger MMP-9 secretion by PSCs and promote tumoral growth. Moreover, this upregulation of MMP-9 secretion appeared specific for HA in PDAC sEVs because the secretion of other cytokines/growth factors investigated was not altered in the absence of HA-low sEVs.

### 2.4. Inhibition of Hexosamine Biosynthetic Pathway Decreased Hyaluronan Levels in PDAC sEVs 

The nutrient-sensing hexosamine biosynthetic pathway (HBP) is a shunt pathway of glycolysis that utilizes glutamine and glucose to make UDP-GlcNAc, the primary substrate for the synthesis of HA [33] (Figure 4A). Since both HBP and HA synthesis are activated in PDAC, using the GEPIA database [34], we analyzed the mRNA expression of the rate-limiting enzyme of HBP glutamine-fructose-6-phospahte amidotransferase 1/2 (GFAT1/2, alias GFPT 1/2) and HA synthesis enzymes hyaluronan synthase-2 (HAS2) and hyaluronan synthase-3 (HAS3) across a set of PDAC patient data. Notably, the dataset displayed significantly higher expressions of GFPT1, GFPT2, HAS2, and HAS3 in PDAC patients (n = 179) compared with healthy controls (n = 171) (Appendix A). Kaplan–Meier survival analysis of PDAC patients (n = 177) in the TCGA dataset using the Kaplan–Meier plotter [35] demonstrated higher levels of GFPT1, GFPT2, HAS2, and HAS3 to be significantly (*p* < 0.05) associated with worse overall patient survival (Figure 4B,C).

It has been demonstrated that HBP is overactivated in PDAC, and activating mutations in the KRAS oncogene (which occurs in> 90% of PDAC cases) dramatically increases glucose uptake and increases flux through the HBP [19,36]. To validate if induction of KRAS mutation alters the intracellular pool of UDP-GlcNAc, an immortalized human pancreatic HPNE cell line engineered to express mutant KRASG12D was subjected to mass spectrometry-based metabolomics analysis. As expected, expression of mutant KRAS in pancreatic ductal cells significantly increased the cellular levels of UDP-GlcNAc and UDP-glucuronic acid (UDP-GlcUA), the two main disaccharides that form the HA glycosaminoglycan polymer (Figure 4D). We next asked if interfering with the HBP pathway represents a potential strategy to eliminate HA in PDAC-derived sEVs. Since GFTA1 activity is dependent on the availability of glutamine, we blocked glutamine utilization using a glutamine analog 6-diazo-5-oxo-L-norleucine (DON). DON in combination with other chemotherapeutic drugs is being evaluated in several cancers [37]. Interestingly, our results showed that treatment of PDAC cell lines ASPC1 and PANC-1 with DON resulted in decreased levels of HA in sEVs isolated from these cells compared with sEVs from cells not exposed to DON (Figure 4E), suggesting that blocking HBP is likely a potential approach to target the deleterious impact of HA in tumor ECM.

## 3. Discussion

Monocytes have emerged as important regulators of cancer development and progression [38]. However, the impact of tumor-derived factors in regulating monocyte subtypes is not well understood. In fact, the monocyte subtypes can contribute to both pro- and antitumoral immunity during cancer progression, and a percentage of monocyte subtypes with protumoral signals could predominate within the tumor microenvironment for cancer cells to evade immune attack [38]. For example, a high proportion of CD14highCD16- classical monocytes predict disease severity in patients with acute pancreatitis [39]. Tumor-derived small extracellular vesicles (sEVs) have been shown to play a significant role in mediating crosstalk between cancer cells and immune cells and reshaping their phenotype to promote cancer progression [40,41,42]. In this study, we demonstrated that sEVs derived from PDAC cells increase the percentage of classical monocyte subtypes (CD14highCD16-) and modulate the expression of HLA-DR and PD-L1. More importantly, the co-expression of HLA-DR and PD-L1 is increased in classical and intermediate monocyte subtypes when exposed to PDAC-derived sEVs. PD-1/PD-L1 axis has become one of the most widely used targets for cancer immunotherapy, and expression of high levels of PD-L1 by cancer cells protects them from escaping immune surveillance [43]. In addition to cancer cells, tumor-infiltrating nonmalignant stromal cells and immune cells, including monocytes, express PD-L1 and contribute to immunosuppression, low survival, and cancer progression [44,45].

Our data indicate that PDAC-derived sEVs modulate circulating monocyte activation and promote an immunosuppressive phenotype before they get recruited to the tumor microenvironment. Tumor-derived sEVs reprogram immune cell phenotypes through the cargo that they deliver to the recipient cells. For example, pancreatic cancer-derived EVs inhibit NK cell function by delivering TGF-β1 to NK cells and inducing Smad2/3 phosphorylation [46]. Another study showed that heat shock protein 72 present on tumor-derived EVs mediates immunosuppressive functions of MDSCs through STAT3 activation [47]. Furthermore, several studies have also reported the role of tumor-EVs in upregulating PD-L1 expression in monocytes [48]. Based on recent reports, the upregulation of PD-L1 in monocytes is achieved through the delivery of specific noncoding RNAs/miRNA by tumor-EVs [49], a mechanism that warrants exploration in the model system used in our study. Our results showed solid evidence of the PDAC-derived sEVs suppressing the monocyte function associated with a reduction in T cell proliferation and an increase in the inflammatory cytokines.

Increased production of glycosaminoglycans (GAG) hyaluronan (HA) by PDAC cells and its abnormal accumulation in the stroma promotes tumor growth and correlates with poor prognosis in PDAC patients [18]. For the first time, we report that the sEVs derived from PDAC cell lines and patient sera are enriched with HA, as was suggested in a previous study, which reported that cells with active HA synthesis make EVs covered with a thick coat of HA [50]. Our findings are significant considering that the accumulation of HA in the stroma is associated with low immune response and poor prognosis in pancreatic cancer [28]. The findings of this study are suggestive of a mechanism by which PDAC sEVs could transport HA to distant sites to create an immunosuppressive pro-tumorigenic niche suitable for tumor cells to grow. This could be a mechanism playing a role in the EVs derived from PDAC cells to create a niche in the liver for the metastatic PDAC cells eliciting macrophage recruitment [6]. The potential for HA to modulate the PDAC TME is also supported by our work identifying the direct effect of sEVs from HA-expressing tumor cells on normal monocytes where HA depletion resulted in a switch from hyperinflammatory (IL-1β) to a chemotactic (MIP1β) cytokine production. Furthermore, monocytes exposed to PDAC sEVs suppressed autologous T cell proliferation; this suppression was reversed when sEVs were depleted of HA. Supporting our hypothesis, Costa-Silva et al. have shown that EVs derived from PDAC cells create a niche in the liver for the metastatic PDAC cells by eliciting macrophage recruitment [6].

Since our study revealed that HA in sEVs enhances the release of MMP-9, known to promote invasion and angiogenesis in PDAC [51,52], we believe that therapeutic targeting of the HBP pathway to downregulate HA levels in PDAC-derived EVs will have significant clinical implications besides potentiating chemotherapeutic efficiency, reported earlier [19]. Our results showed that inhibiting the rate-limiting enzyme GFAT1 of HBP significantly decreased HA levels in PDAC sEVs. GFAT is often upregulated in cancer and correlates with cellular UDP-GlcNAc content and tumor HA levels [33]. GFAT expression also correlates with the expression of HA synthase enzyme (HAS) and their co-expression is associated with poorer overall cancer patient survival compared with those expressing these enzymes alone [33]. In line with this, we found that the expression of GFAT and HAS is highly upregulated in PDAC patients compared with healthy controls, suggesting a link between HBP and HA production in PDAC. Supporting this notion, a recent study showed that HBP was overactivated in PDAC, and inhibition of GFAT1 decreased HA levels within the tumor microenvironment [19]. This also led to the remodeling of the immune landscape by increasing the infiltration of both CD68+ macrophages with antitumor activity and cytotoxic CD8+ T cells [19]. It is relevant in this context that alterations in the HBP pathway have been shown to regulate dynamic cellular events involving hyaluronan production associated with cancer progression [33].

The dynamic bidirectional communications between PDAC cells and PSCs promote the activation of PSC and the secretion of different growth factors/cytokines necessary for PDAC progression [53]. Studies have also reported the role of PSCs in creating an immunosuppressive microenvironment by releasing suppressive cytokines/growth factors such as IL-6, IL-10, CXCL12 (SDF), MCP-1, TGF-β, GM-CSF, PGE2, and VEGF, which contribute to T cell exhaustion and dysfunction [54,55,56,57]. Importantly our findings that PDAC sEVs promote the release of various cytokines by PSCs, including IL-6, MCP-1, GM-CSF, and VEGF, underscore an underlying indirect mechanism for sEVs, acting via regulating the functional state of PSCs, in driving progression and immune evasion in PDAC.

## 4. Materials and Methods

### 4.1. Cell Lines

Human pancreatic adenocarcinoma cell lines AsPC1 and PANC-1 were obtained from ATCC. Additional human pancreatic adenocarcinoma cell lines HPAF II, HuP-T3, and PA16C were provided by Dr. Anirban Maitra. Human pancreatic stellate cells from primary immortalized cell lines were kindly provided by Dr. Rosa Hwang from the Department of Surgical Oncology, MDACC. Two normal pancreatic cell lines were included as controls: a normal human pancreatic ductal epithelial cell line (HPDE) and a normal, hTERT-immortalized pancreatic duct epithelial cell (HPNE).

Cell lines were cultured in an appropriate medium supplemented with 10% fetal bovine serum, 1% penicillin/streptomycin, and glutamine. Peripheral blood mononuclear cells (PBMCs) were isolated from healthy donor buffy coats. Monocytes were isolated by MACS (Miltenyi Biotech) using negative and CD14+ selection according to the manufacturer’s instructions. Cell cultures were maintained at 37 °C in a humidified atmosphere containing 5% CO2. Cells were routinely screened for mycoplasma contamination.

### 4.2. Patient Samples

Clinical information and serum samples collected from PDAC patients, after informed consent, were obtained from the pancreatic tissue bank of MD Anderson Cancer Center (Appendix A). Patient samples used in this study were followed at MD Anderson Cancer Center. Serum was separated from whole blood samples, snap-frozen and stored at −20 °C.

### 4.3. Isolation and Characterization of Small Extracellular Vesicles

Serum-free conditioned medium of cell lines harvested after 48 h were centrifuged at 2000× *g* for 30 min to remove cells and debris. The cleared supernatant was then concentrated using a 100,000 MW cutoff centrifugal concentrator (corning) to enrich the sEV fraction. The retentate was then mixed well with 0.5 volumes of total sEV isolation reagent (Thermo Fisher Scientific, Waltham, MA, USA) and incubated overnight at 4 °C. Samples were then centrifuged at 10,000× *g* for 1 h at 4 °C. Supernatants were discarded and sEV pellets at the bottom of the tubes were resuspended in 1X PBS.

sEVs in human serum were isolated using size exclusion-based chromatography (System Bioscience, Palo Alto, CA, USA) according to the manufacturer’s instructions. Briefly, 250 µL of serum was loaded to prewashed columns, incubated for 30 min at room temperature, and centrifuged to elute the sEVs. Isolated sEVs were stored at 4 °C for immediate use or at −80 °C for long-period storage.

The sEV size distribution and concentration were measured by nanoparticle tracking analysis by a ZetaView instrument. With nanoparticle tracking analysis, each individual particle in the field of view was counted and tracked in short video clips, creating accurate concentration calculations and particle size distributions.

### 4.4. Western Blotting

sEVs were characterized for the presence of exosome-associated marker proteins CD63 and Alix using Western blotting. Briefly, sEVs were lysed using RIPA buffer in ice for 10 min. Lysates were loaded to SDS-PAGE gels and then transferred onto 0.22 um polyvinylidene difluoride membrane (Millipore, Burlington, MA, USA). Membranes were blocked and incubated with primary antibodies at 4 °C overnight, followed by incubation with the horseradish peroxidase-conjugated secondary antibody at room temperature. Finally, the membranes were visualized with ECL substrates (Thermo Fisher Scientific).

### 4.5. Monocyte Isolation from PBMCs

Monocytes were enriched from thawed PBMCs from 7 normal donors with magnetic beads using human pan monocyte isolation human kit (Miltenyi Biotec, Gaithersburg, MD, USA) (Cat. 130-096-537) with negative selection. The thawed PBMCs were washed with serum-free RPMI 1640 with L-glutamine (Corning, VA, USA) before enrichment. Monocyte yield and viability were assessed using a K2 cell counter (Nexcelom Biosciences, Lawrence, MA, USA). Following isolation, the cells were cultured with RPMI 1640 with L-glutamine and 5% of FBS at 37 °C 5% CO2. Purity was evaluated by flow cytometry and monocytes were identified by CD14+CD3-CD56-CD19-CD20- (Appendix A). The subpopulations were analyzed by the expression of CD14+ (LPS co-receptor) and CD16+ (FCRγ−I) as classical (CD14highCD16-), intermediate (CD14highCD16low), and nonclassical monocytes (CD14lowCD16high).

### 4.6. Treatment of Monocytes with EVs Derivate from Pancreatic Cell Lines

Cells were rested in RPMI (Gibco, Vacaville, CA, USA) containing 5% FBS at 37 °C 5% CO2 for 6–8 h after enrichment. After resting, the monocytes were co-cultured with EVs from pancreatic cell lines ASPC-1, PANC-1, HPAF-II, and PANC-1 HA-low in a ratio 1:20 (monocyte:EV) and incubated overnight at 37 °C, 5% CO2 with RPMI culture media without SFB (Gibco). The cells were harvested, washed, and stained using flow cytometry.

### 4.7. Flow Cytometry Analysis

The subpopulations were analyzed using an established monocyte flow cytometry panel anti-CD14 BV421 (clone MoP9, BD), anti-CD16 BV711 (clone 3G8, BD), anti-PD-L1 BV650 (clone MIH1, BD), anti-CD86 APC-R700 (clone 2331 FUN-1, BD), anti-HLA-DR BUV395 (clone G46-6, BD), and anti-CD64 FITC (clone 22, Beckman Coulter, Brea, CA, USA). Samples were acquired on a BD Fortessa X20, and data were analyzed using FlowJo software (v10.7.1). Analysis was performed following doublet exclusion and subsequent viability using a live/dead fixable yellow dye from Fisher L-34968. Samples were acquired within 24 h of staining. Fluorescence minus one control was used to determine positive and negative gates. Subgating was only performed on parental populations containing more than 100 events for data QC. All the fold changes were calculated with the non-stimulated (NS) healthy donor control.

### 4.8. Cytokine Quantification from Supernatants and Cell Lysates

Enriched monocytes from 3 healthy independent normal donors’ PBMCs were seeded at 1 × 10^5^ cells in 96-well plates and rested with 200 μL of fresh RPMI 1640 with L-glutamine and 5% of FBS at 37 °C 5% CO2 during 6–8 h prior to contact with sEVs. The purity of monocyte subpopulation per donor was analyzed by flow cytometry previously described. Following incubation, the media were retired and refreshed with RPMI 1640 with serum-free L-glutamine. sEVs from pancreatic cancer cell lines ASPC1, PANC-1, and PANC-1 HA-low, two normal pancreatic cell lines: human pancreatic ductal epithelial cell line (HPDE) and hTERT-immortalized pancreatic duct epithelial cell (HPNE) were added to the monocytes in a 1:2 ratio or without EVs as non-stimulated control (NS). The stimulation conditions were 37 °C, 5% CO2 for 12 h. Supernatants and monocytes were recovered and treated with lysing buffer: T-PER tissue protein extraction buffer (ThermoFisher), 5 M sodium chloride with cOmplete protease inhibitor cocktail, and PhosSTOP phosphatase inhibitor 1x (Roche), vortex 2 min at RT. Immediately and at 4 °C, the lysate was centrifuged, and the supernatant was recovered for protein quantification by Pierce BCA Protein Assay Kit (ThermoFisher) in duplicate. Cytokine quantification was performed using a 65-plex immunomonitoring panel. Protein concentration was normalized to 1 mg/mL and analyzed in triplicate. The data acquisition and analysis were performed using XPotent 4.2 for Luminex and Bio-Plex Manager Software 6.1 (Bio-Rad, Hercules, CA, USA). For the graphics and statistical analysis, we used GraphPad Prism 9. The statistical analyses conducted were one-way ANOVA, post hoc Tukey’s multiple comparison test, and alpha threshold 0.05 (95% confidence interval). * *p* < 0.05, ** *p* < 0.01, *** *p* < 0.001, **** *p* < 0.0001.

### 4.9. Proliferation of T Cell Inhibition

Enriched monocytes from 4 healthy independent normal donors’ PBMCs were seeded 1 × 10^5^ in 96-well plates and rested with 200 μL fresh RPMI 1640 with L-glutamine and 5% of FBS at 37 °C 5% CO2 during 6–8 h prior to contact with EVs. The purity of monocyte subpopulation per donor was analyzed by flow cytometry as described in the main text and was used for calculating the monocyte:T cell ratio. Following incubation, the media were retired and refreshed with RPMI 1640 with serum-free L-glutamine and treated overnight with the sEVs from pancreatic cancer or normal cell lines and without sEVs as non-stimulated control (NS). T cells from autologous normal healthy donors, enriched from PBMCs by nonadherence portion in 1 h culture in RPMI 1640 with L-glutamine and 5% of FBS at 37 °C and 5% CO2 on a dish plate, were stained with 5 mM carboxyfluorescein succinimidyl ester (CFSE) for 15 min at 37 °C, washed twice with 1X PBS, and resuspended fresh RPMI-FBS. Stained T cells and monocytes were seeded in a 1:5 ratio (monocyte: T cell) and incubated for 5 days with anti-CD3 antibody clone OKT3 (GMP-grade, 30 ng/mL), Miltenyi #170-076-124 and Proleukin^®^ recombinant human IL-2 (6000 IU/mL), Prometheus Laboratories. Finally, cells were harvested, washed, and analyzed by flow cytometry. Analysis was performed using FlowJo-10.9.0. The T cell area for proliferation was selected by forward and side scatter and singlet gates. The FlowJo proliferation tool permitted the determination of the undivided area with the control tube of T cells stained with CFSE alone (no stimulation, no monocytes). All the index values represented the division, expansion, and percent of divided cells provided by the proliferation tool algorithm. The percent divided by generations was the full percentage of the positive proliferation area in the histograms. For the graphics and statistical analysis, we used GraphPad Prism 9, and the statistics conducted were one-way ANOVA, post hoc Tukey’s multiple comparison test, and alpha threshold 0.05 (95% confidence interval). * *p* <0.05, ** *p* < 0.01, *** *p* < 0.001, **** *p* < 0.0001.

### 4.10. Analysis of Hyaluronan

ELISA was utilized to quantify HA (R&D Systems) in sEVs, following the manufacturer’s instructions. The quantity of exosomes was determined by BCA assay (Thermo Scientific) for the measurement of total protein. For some experiments, cells were treated with 0.5 mM 4-MU to deplete HA levels in sEVs.

### 4.11. Transmission Electron Microscopy

sEV samples were placed on 100 mesh carbon-coated, formvar-coated copper grids treated with poly-l-lysine for approximately 1 h. Samples were then negatively stained with Millipore-filtered aqueous 1% uranyl acetate for 1 min. The stain was blotted dry from the grids with filter paper and samples were allowed to dry. Samples were then examined in a JEM 1010 transmission electron microscope (JEOL, USA, Inc., Peabody, MA, USA) at an accelerating voltage of 80 KV. Digital images were obtained using the AMT imaging system (Advanced Microscopy Techniques Corp., Danvers, MA, USA).

### 4.12. Dot Blot Analysis

sEVs derived from pancreatic cancer cell lines or patient serum were spotted on a strip of nitrocellulose membrane, which was then dried at room temperature for 1 h. Bovine serum albumin (BSA) was used as a negative control. The membrane was then blocked with 5% BSA/TBST for 1 h at RT, followed by overnight incubation with biotinylated hyaluronic acid binding protein (from Sigma, St. Louis, MO, USA) in 3% BSA/TBST at 4C. After 3 washes with TBST, the membrane was incubated with HRP-conjugated streptavidin in 3% BSA/TBST for 1 h at room temperature. Following 3 washes with TBST, the signals were visualized using chemiluminescence.

### 4.13. Human Cytokine Array

The proteome profiler human XL cytokine array was purchased from R&D Systems (Cat # ARY022B). This membrane-based antibody array allows for the parallel determination of the relative levels of selected human cytokines and chemokines. Array membranes were incubated for 24 h with serum-free conditioned medium from human pancreatic stellate cells that were either treated with HA-high or HA-low sEVs. Membranes were then processed as recommended by the manufacturer’s instructions and signals were visualized using chemiluminescence. Signal densities of two capture spots per cytokine on developed X-ray film were scanned and quantitated using ImageJ software, version 1.53t.

### 4.14. Targeted Metabolomics Using Mass Spectrometry

Samples were prepared and analyzed for liquid chromatography-mass spectrometry (LC-MS)-based metabolomics using single reaction monitoring (SRM) as previously reported [58]. Briefly, frozen cell pellets stored at −80 °C were thawed at 4 °C and subjected to three freeze–thaw cycles in liquid nitrogen before the addition of 750 µL of ice-cold methanol:water (4:1) containing 20 µL of spiked internal standard to each cell extract. Ice-cold chloroform and then water were sequentially added in a 3:1 ratio to each cell extract. Organic and aqueous layers were isolated and then combined before deproteinization using a 3 kDa Amicon Ultracel molecular filter (Millipore, Billerica, MA, USA). Filtrates were dried in a vacuum (Genevac EZ-2plus, Gardinier, NY, USA) and then resuspended in 1:1 methanol:water with 0.1% formic acid for LC-MS analysis. The metabolomics data from each peak were analyzed using Agilent Mass Hunter Quantitative Analysis software, version 10. The peak area was normalized to internal standards and log2-transformed. The heat map was generated using the R statistical system.

### 4.15. Microarray Dataset Analyses

The pancreatic cancer dataset from the GEPIA database (34) was used to explore the mRNA expression of GFPT1, GFPT2, HAS2, and HAS3 in PDAC patients (n = 179) vs. healthy controls (n = 171). Kaplan–Meier survival analysis of PDAC patients (n = 177) with high and low expression of these genes was performed using the TCGA dataset and Kaplan–Meier plotter (35).

### 4.16. Statistical Analysis

Flow cytometry statistical significances were evaluated by analyzing the SD from seven independent experiments using the ANOVA test and Dunn’s multiple comparison test, and a *p*-value of less than 0.05 was considered statistically significant. All group numbers and explanations of significant values are presented within the figure legends. Statistical analysis of HA in patients post- and pretreatment was performed by paired t-test for surgical specimens within each group and Mann–Whitney test for differential group comparisons. Heatmaps represent the median of the normalized data from each dataset. Single linear regression for correlation analysis using GraphPad Prism v9 was used for statistical analysis comparing HLA-DR and PD-L1 in HA and HA-low.

## 5. Conclusions

sEVs generated from pancreatic cancer cells exhibited a significant immunomodulatory role that preferentially influences the activation status of monocyte subtypes. Monocytes exposed to sEVs underwent distinct phenotypic changes with specific expression of activation and inhibitory receptors along with antigen presentation machinery. The presence of HA in the sEVs had little effect on regulating the proportions of monocyte subtypes, while HA in PDAC sEVs enhanced monocyte production of IL-1β, influenced their ability to inhibit T cell proliferation, and enhanced MMP-9 production from stellate cells. Interruption of the hexosamine biosynthetic pathway in pancreatic cancer cells represents a potential approach for targeting the adverse impact of HA in PDAC. Further research using humanized animal models and clinical sample validation may uncover additional mechanisms through which sEVs drive a suppressive environment in PDAC.

## Figures and Tables

**Figure 1 ijms-24-14652-f001:**
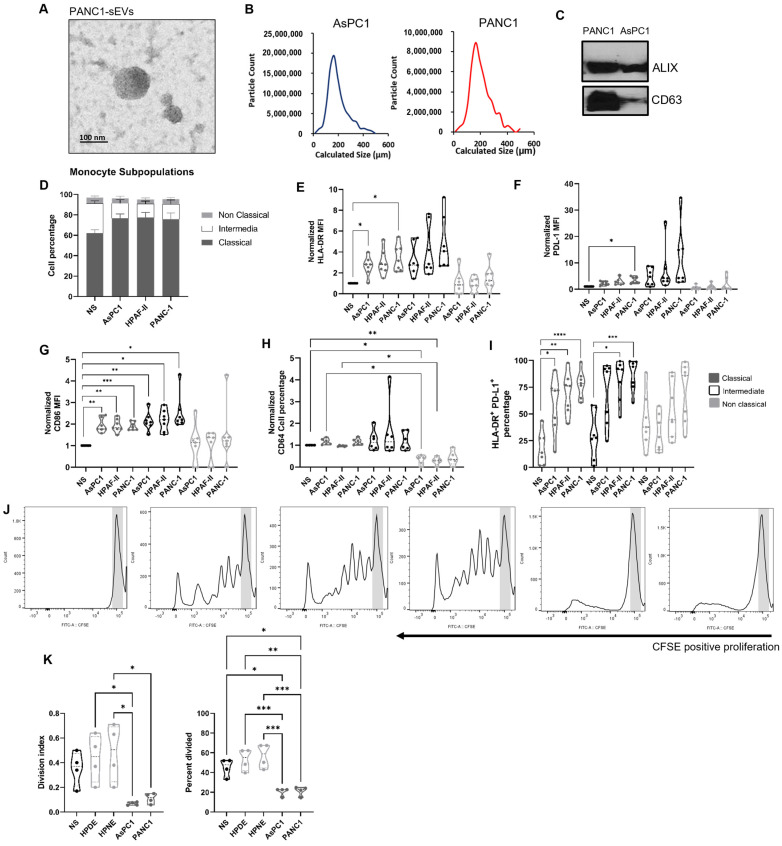
Pancreatic cancer-derived small extracellular vesicles (sEVs) preferentially modulate the monocyte landscape. Characterization of PDAC-derived sEVs by (**A**) transmission electron microscope, (**B**) nanoparticle tracking analysis by ZetaView, and (**C**) by Western blot. (**D**) Monocyte subpopulation percentages, enriched from PBMCs healthy donors and exposed to sEVs isolated from human PDAC cells AsPC1, HPAF-II, and PANC-1 analyzed by flow cytometry. NS: non-stimulated with sEVs. Graphs show ± SD from seven independent experiments; statistical significance was assessed using the ANOVA test and Tukey’s multiple comparison test, *p*-value * < 0.05, ** < 0.01, *** < 0.001, and **** < 0.0001. (**E**–**I**) The same color pattern was used to identify monocyte subpopulations in all the graphs. Normalized expression markers analysis (median fluorescence intensity MFI or percentage) on monocyte subpopulations by flow cytometry. (**J**) Representative plots from one normal donor T cell proliferation assay with autologous cells. The area in gray is representative of undivided cells. (**K**) Division index and percent of cells divided; all data were obtained with proliferation analysis tool from FlowJo.

**Figure 2 ijms-24-14652-f002:**
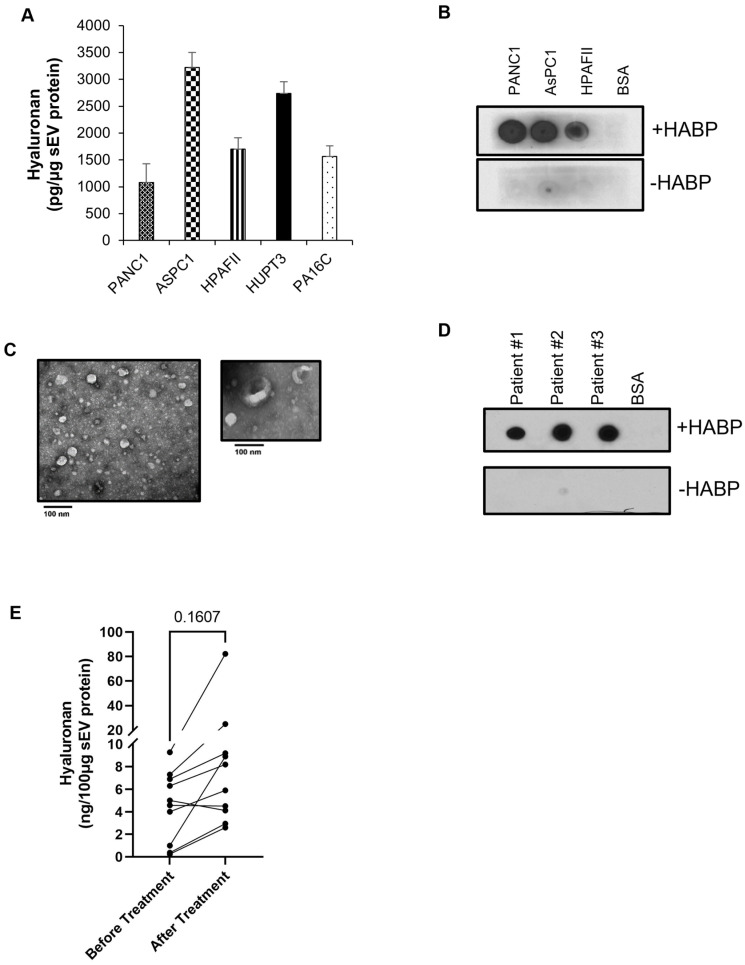
sEVs derived from PDAC patients enriched with hyaluronic acid. (**A**) ELISA quantification of hyaluronan in sEVs isolated from human PDAC cell lines. Results are mean values from three different sEV preparations ± SD. (**B**) Detection of hyaluronan in sEVs by dot blot analysis. PDAC cell line-derived sEVs were spotted on nitrocellulose membrane and incubated overnight with or without hyaluronan binding protein (HABP). Signals were visualized by chemiluminescence. BSA serves as the negative control. (**C**) sEVs isolated from the serum of PDAC patients, using size exclusion-based chromatography, were examined by transmission electron microscopy. Note that the size (bar = 100 nm) and shape (round morphology) are consistent with their identity as sEVs. The image in the insert reveals a “cup shape” morphology, which is typical of sEVs called exosomes. (**D**) Dot blot analysis or (**E**) ELISA quantification of hyaluronan in sEVs isolated from the serum of 10 PDAC patients before any treatment or after subsequent rounds of treatment. Statistical significance was assessed using a paired *t*-test.

**Figure 3 ijms-24-14652-f003:**
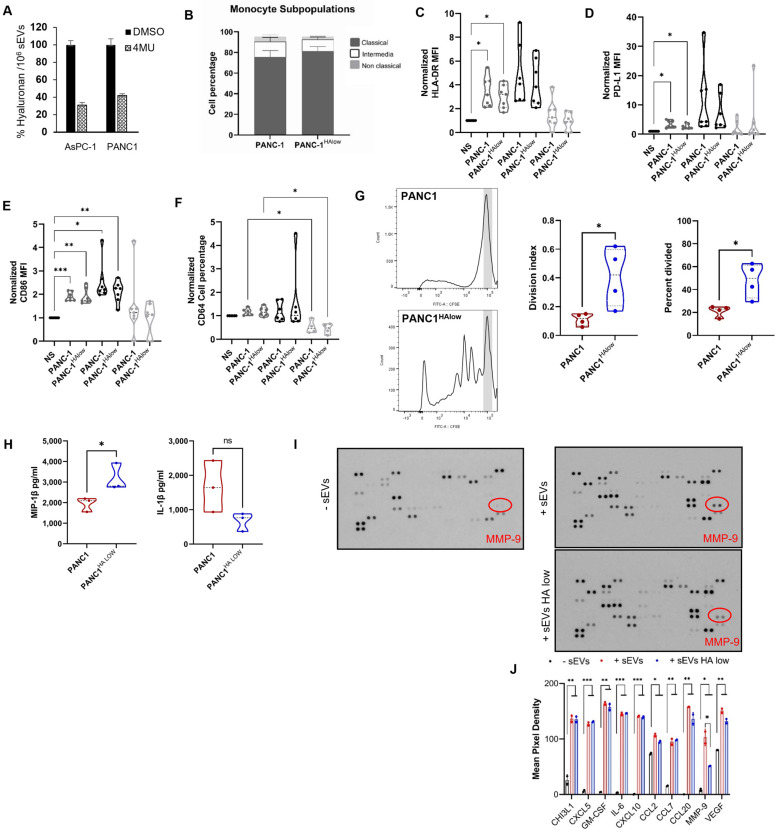
PDAC sEVs can modulate marker expression on monocyte subpopulations and induce increased soluble factors in stellate cells. (**A**) ELISA quantification of hyaluronan in conditioned media of AsPC1 and PANC-1 pancreatic cancer cell lines treated with 0.5 mM 4-methylumbelliferone (4MU) or DMSO (as control) (n = 3). (**B**) Monocyte subpopulation percentages exposed to sEVs isolated from PANC-1 cells either treated with DMSO or 4-MU (HA-low) analyzed by flow cytometry. (**C**–**F**) Normalized expression markers analysis (median fluorescence intensity MFI or percentage) on monocyte subpopulations co-incubated with PANC-1 sEVs (DMSO control) or PANC-1 HA-low sEVs (4-MU) by flow cytometry. NS: non-stimulated with sEVs. Graphs show ± SD from seven independent experiments; statistical significance was assessed using the ANOVA test and Tukey’s multiple comparison test, *p*-value * < 0.05, ** < 0.01, and *** < 0.001. (**G**) Representative T cell proliferation plot from one normal donor and division index and percent of cells divided values obtained from the FlowJo proliferation tool, the gray area is representative of undivided cells. (**H**) Cytokine production from three normal donors’ monocytes after exposure to sEVs with or without HA. (**I**) A representative antibody array that simultaneously detects 105 different human cytokines/chemokines was utilized to determine the sEVs-induced secretome of human pancreatic stellate cells. Array membranes were incubated with serum-free conditioned media of human pancreatic stellate cells treated with PANC-1 or PANC-1 HA-low sEVs. Duplicate dots at the three corners of the membrane represent the reference spots. Dots with a red circle represent MMP-9. (**J**) Different cytokines/chemokines secreted by pancreatic stellate cells treated with PDAC-derived sEVs (**middle** array) compared with sEVs untreated cells (**left**).

**Figure 4 ijms-24-14652-f004:**
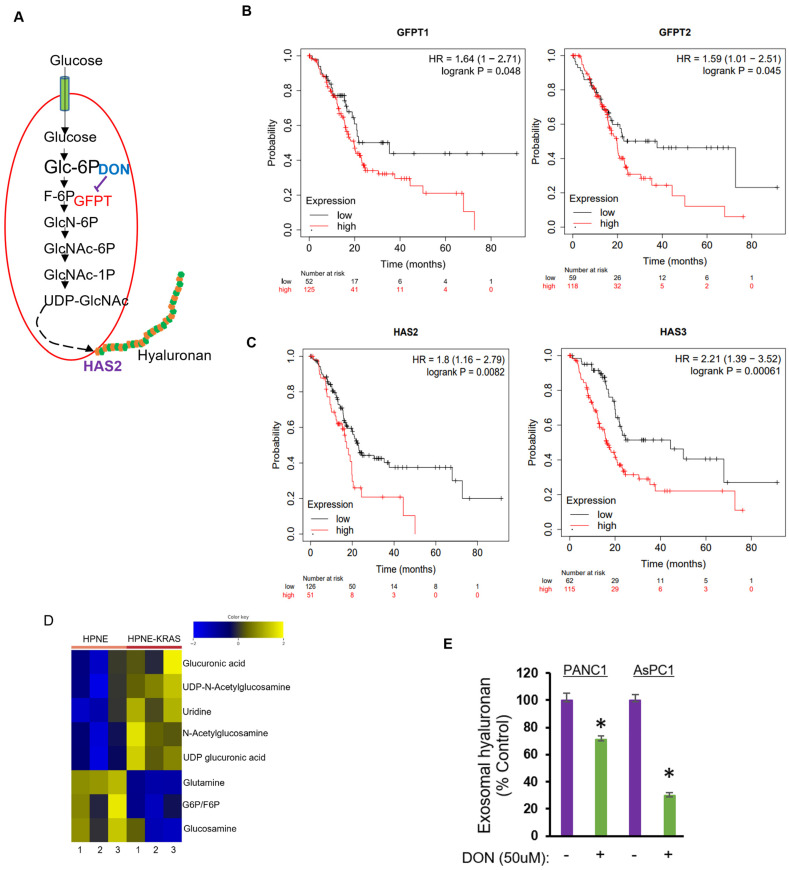
Targeting the hexosamine biosynthetic pathway attenuates the levels of HA in PDAC sEVs. (**A**) Schematic overview of the hexosamine biosynthetic pathway. Glc-6P, glucose-6-phaosphate; F-6P, fructose-6-phaosphate; GlcN-6P, glucosamine-6-phosphate; GlcNAc-6P, N-acetylglucosamine-6-phosphate; GlcNAc-1P, N-acetylglucosamine-1-phosphate; UDP-GlcNAc, uridine diphosphate N-acetyl glucosamine; GFPT, glutamine: fructose-6-phaosphate transaminase; HAS2, hyaluronan synthase-2; DON, 6-diazo-5-oxo-L-norieucine. (**B**,**C**) Kaplan–Meier overall survival curves of patients with PDAC (n = 177) and either with high or low expression of GFPT1, GFPT2, HAS2, and HAS3. Elevated levels of GFPT1, GFPT2, HAS2, and HAS3 significantly (*p* < 0.05) associated with worse overall patient survival. (**D**) Heat map of eight different metabolites in immortalized pancreatic ductal HPNE cells with or without G12D mutation of KRAS gene. Rows represent the distinct metabolites and columns represent the triplicate analysis of the two different cells. Shades of yellow and blue represent higher and lower levels of metabolites, relative to the median metabolite levels. (**E**) ELISA quantification of hyaluronan in sEVs from PANC-1 and AsPC1 cells treated with or without the GFPT1 inhibitor DON (50 µM). Results are mean values from three independent experiments ± SD. * *p* < 0.05 vs. DON untreated cells.

## Data Availability

Data and materials can be made available upon request to the co-corresponding authors.

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
