# Peer review of "Differential Effects of Pancreatic Cancer-Derived Extracellular Vesicles Driving a Suppressive Environment"

_ijms, 2023, doi:10.3390/ijms241914652_

Round 1
Reviewer 1 Report
Purushothaman et al checked the impact of small EVs on monocyte lineage transition as well as expression of checkpoint receptors in Pancreatic Cancer. The work is very interesting for the researchers working in the field of pancreatic cancers. I have few concerns, that can be worth and should be incorporated.
1. I haven’t find any concluding remarks for the work. It should be worth, if authors include one separate section as conclusion.
2. Since the authors covered EVs, that includes both exosomes (30nm to 150nm) and microvesicles (50-1000nm). Most of the EVs markers, used here, just for exosomes. Has authors characterized microvesicles?. Because, some EVs especially in the range of 50-150nm belongs to both microvesicles and exosomes.
3. The term "mechanisms underlying tumor-immune cell cross-talk" is somewhat technical. Consider providing a brief explanation or simplifying the language to enhance readability.
4. When referring to MISEV2018, it would be beneficial to provide a brief explanation of what it is for readers, who may not be familiar with this acronym.
5. While line 61-69 mentions various immune cells and their roles, it might benefit from providing a bit more context or background information for readers who may not be familiar with these terms. Explain why these immune cells are relevant in the context of pancreatic cancer.
6. While it's common to use abbreviations for term HLA-DR, make sure to introduce this abbreviation with their full forms the first time they are used in the paragraph to help readers understand them.
7. Overall, the work is very interesting.
Minor English editing is required.
Author Response
Differential effects of pancreatic cancer-derived extracellular vesicles driving a suppressive environment
Reviewers responses
Reviewer 1
Purushothaman et al checked the impact of small EVs on monocyte lineage transition as well as expression of checkpoint receptors in Pancreatic Cancer. The work is very interesting for the researchers working in the field of pancreatic cancers. I have few concerns, that can be worth and should be incorporated.
- I haven’t found any concluding remarks for the work. It should be worth, if authors include one separate section as conclusion.
Response: We thank the reviewer, and a conclusion has been added in a separated section.
- Since the authors covered EVs, that includes both exosomes (30nm to 150nm) and microvesicles (50-1000nm). Most of the EVs markers, used here, just for exosomes. Has authors characterized microvesicles? Because, some EVs especially in the range of 50-150nm belongs to both microvesicles and exosomes.
Response: Thank you for this important suggestion. Based on the size distribution, the particles used in this study includes both exosomes and microvesicles. In recognition of this fact, we refer to the particles as small Extracellular Vesicles (sEVs) and agree with the reviewer that the markers used to characterize EVs are prototypical exosome markers, which we have mentioned in line 146 of page 3 and line 493 of page 14.
- The term "mechanisms underlying tumour-immune cell cross-talk" is somewhat technical. Consider providing a brief explanation or simplifying the language to enhance readability.
Response: We thank and agree with the reviewer and the language of the sentence was changed to: “In depth understanding of the fundamental mechanisms of the tumor and immune cell communication is necessary…”
- When referring to MISEV2018, it would be beneficial to provide a brief explanation of what it is for readers, who may not be familiar with this acronym.
Response: MISEV2018 is now described in page 2, line 57-59
- While line 61-69 mentions various immune cells and their roles, it might benefit from providing a bit more context or background information for readers who may not be familiar with these terms. Explain why these immune cells are relevant in the context of pancreatic cancer.
Response: We thank the reviewer for the suggestion. A simplified explanation for the cells in the context of PDAC was added to the manuscript.
- While it's common to use abbreviations for term HLA-DR, make sure to introduce this abbreviation with their full forms the first time they are used in the paragraph to help readers understand them.
Response: We are thankful for the observation; the introduction of the abbreviation was made on line 72-73.
- Overall, the work is very interesting.
Response: We thank the reviewer for the comments and suggestions for improvement of our manuscript.
Reviewer 2 Report
The authors exposed small extracellular vesicles (sEVs) from pancreatic ductal adenocarcinoma (PDAC) cells to peripheral blood monocytes from healthy donors to elucidate their role in intercellular crosstalk and tumor progression. They also investigated the effects of hyaluronic acid (HA) from sEVs to fibroblasts, monocytes and pancreatic stellate cells PSC). Various lines of evidence support the idea that sEVs affect tumor progression in several ways.
Overall the experiments are well described and the results and discussion appear to the reviewer very well balanced, although there are so many aspects of CD44 (and splice variants) and hyaluronate that not everything can or should be covered.
The reviewer thinks this is a publishable set of findings suitable for the journal.
Author Response
Differential effects of pancreatic cancer-derived extracellular vesicles driving a suppressive environment
Reviewers responses
Reviewer 2
The authors exposed small extracellular vesicles (sEVs) from pancreatic ductal adenocarcinoma (PDAC) cells to peripheral blood monocytes from healthy donors to elucidate their role in intercellular crosstalk and tumor progression. They also investigated the effects of hyaluronic acid (HA) from sEVs to fibroblasts, monocytes and pancreatic stellate cells PSC). Various lines of evidence support the idea that sEVs affect tumor progression in several ways.
Overall the experiments are well described and the results and discussion appear to the reviewer very well balanced, although there are so many aspects of CD44 (and splice variants) and hyaluronate that not everything can or should be covered.
The reviewer thinks this is a publishable set of findings suitable for the journal.
Response: We sincerely thank the reviewer for the positive comments about our study.